
**Multi-coverage Optimal Location Model for Emergency Medical**
**Services (EMS) facilities under various disaster scenarios:A case study**
**of urban fluvial floods in the Minhang District of Shanghai, China**
Yang Yuhan[a,b], Yin Jie[a,b,c]
a. Key laboratory of Geographic Information Science (Ministry of Education), East China Normal University, China
b. School of Geographic Sciences, East China Normal University, China
c. Institute of Eco-Chongming, East China Normal University, China
*Correspondence to:* Yin Jie (rjay9@126.com)
**Abstract:** Emergency medical services (EMS) response is extremely critical for pre-hospital lifesaving when disaster events occur.
However, disasters increase the difficulty of rescue and add valuable minutes to travel times between dispatch and arrival, thereby
increasing the pressure on emergency facilities. Hence, facility location decisions play a crucial role in improving the efficiency of rescue
and service capacity. In order to avoid the failure of EMS facilities during disasters and meet the multiple requirements of demand points,
we propose a multi-coverage optimal location model for EMS facilities based on the results of disaster impact simulation and prediction.
To verify this model, we explicitly simulated the impacts of fluvial flooding events using the FloodMap model. The simulation results
suggested that even low-magnitude fluvial flood events resulted in a decrease in the EMS response area. The integration of the model
results with a Geographical Information System (GIS) analysis indicated that the optimization of the EMS locations reduced the delay
in emergency responses caused by disasters and significantly increased the number of rescued people and the coverage of demand points.
**Keywords**: Disaster events; Emergency Medical Service; Multi-coverage Location model; Scenario simulation

1. Introduction

Urban disasters represent a serious and growing challenge. Against the backdrop of urbanization, demographic
growth, and climate change, the causes of disasters are changing and their impacts are increasing. Both natural
hazards such as flash flooding and human-caused accidents such as fires threaten and induce panic in people and
cause casualties and property loss(Kates et al., 2001; Makowski and Nakayama, 2001). In order to deal with
emergencies effectively, a large number of emergency service facilities may be called upon simultaneously. However,
the demands being placed upon emergency services often exceed the resources made available by governments(Liu
et al., 2017). Therefore, the maintenance of efficiency and quality of emergency services during disasters is the key
to emergency management. A scientific and pragmatic approach to the choice of locations and allocations of
emergency service facilities reduces traffic congestion and the risk of secondary incidents during an emergency,
which, in turn, reduces transport costs and greatly improves the efficiency of rescue services.

Over the last few decades, research into traditional location theory has resulted in a number of models to determine



the optimal location of emergency services; the most commonly used models are the P-center model (Hakimi, 1964),
the P-median model (Hakimi, 1965), and the covering model (Brandeau and Chiu, 1989). Among these models, the
covering model is the most widely investigated and applied model; the objective of the model is to improve the
coverage of facilities within a limited time or distance to meet the service requirements (Ge and Wang et al., 2011).
The most common covering models are the Location Set Covering Model (LSCM) (Toregas and Revelle, 1972) and
the Maximum Covering Location Problem (MCLP) model (Church and Revelle, 1974). The focus of the LSCM is to
minimize the number of facilities needed to cover all demand points but it has been shown to lead to an unequal
allocation of facilities or a large increase in costs. Due to these limitations, the MCLP model was developed to ensure
that existing emergency facilities were used to obtain the maximum coverage of the demand points. Drawing upon
the LSCM and MCLP model, a number of researchers have optimized the associated algorithms in terms of facility
workload limits(Pirkul and Schilling, 1991), cost(Su et al., 2015), and level of coverage(Gendreau and Laporte, 1997)
to solve various practical problems or achieve rescue objectives. Other types of models are suitable for location
decision problems that do not include time or distance restrictions, such as the P-center model and the P-median
model, P means the number of facilities that need to be built. The P-center model mainly considers equitable service;
it selects P facilities by minimizing the maximum distance between the demand points and the facilities. The P-
median model not only takes into account the efficiency of the emergency facilities but it also minimizes the sum of
the weighted distance between the demand points and the P facilities (Chen and You, 2006).
All of the above models are static in the sense that they do not consider uncertainties in the emergency service process.
For example, large-scale emergencies are likely to require high levels of healthcare to the extent that emergency
service facilities would need to provide transportation to other facilities that are beyond the immediate area.
Furthermore, the limited ambulance resources at any one emergency station would restrict the capacity of the
emergency medical service (EMS) when multiple demand points make simultaneous requests. Any further demands
placed upon the emergency services would cause them to fail, resulting in potential loss of life. To minimize these
fluctuations in an EMS system, approaches have been proposed that involve multi-coverage models (Moeini and
Jemai, 2015). In 1981, Daskin and Stern(1981) put forward their hierarchical objective set covering model (HOSC),
in which they introduced the concept of 'multiple coverage of zones'; the objective was to minimize the number of
necessary facilities such that the demand was still met and to maximize the coverage of the demand points. However,
HOSC had one major shortcoming; it potentially led to the congestion of emergency vehicles. To solve these problems,
Hogan and ReVelle (1986) proposed an alternative approach to coverage in their maximal backup coverage models
BACOP1 & BACOP2. These models cover each demand point at least once but the multiple coverage of different
demand points with the same coverage level resulted in a waste of vehicles resources (Ge and Wang et al., 2011).
Considering that there is usually a limited financial budget for the provision of emergency services, it is not feasible
to cover all demand points multiple times.
The aforementioned traditional location models ignored the impacts of specific disasters but we suggest that these
impacts must be part of any decision regarding the location of emergency services. Apart from causing casualties, a
disaster may also damage emergency facilities; furthermore, damage to buildings and roads will lead to traffic
congestion and render emergency rescue more difficult than usual. To avoid these problems, research has been
conducted on choosing the location of emergency service facilities in response to large-scale emergencies. Jia and
Ordóñez (2007) defined the main characteristics of ideal locations of emergency service facilities as "timeliness",
"fairness", and "resistance to failure". Chen and You (2006) established a multi-objective decision model for the
location of emergency rescue facilities by integrating the MCLP model, the P-median model, and the P-center model.
In this integrated model (which focused on large-scale disasters), emergency facilities were allocated using different



strategies. Jia and Ordóñez (2007) investigated models for EMS facility location in response to disasters and
compared three heuristic algorithms (genetic algorithm, location-allocation algorithm, and Lagrange relaxation
algorithm) applicable to emergency scenarios and location models.
Having taken account the results of previous studies, here we describe a novel approach for the optimization of EMS
efficiency under various disaster scenarios. We propose a multi-coverage optimal location model, whose output
depends on the impact of a disaster and the levels of demand made on the EMSs. We use a scenario-based method
and Geographical Information System (GIS)-based network analysis to quantify the impacts of a disaster on the urban
EMS response. The coverage level of the demand points is determined by the population, the EMS calls for help, and
other factors that reflect the demand level of the demand points; these factors determine how often the demand point
needs to be covered by emergency facilities within a predefined time. The higher the demand coverage level, the
more often a demand point needs to be covered by the service area of the emergency facilities in a given time period.
The main purpose of our location model is to reduce the probability of delays in the emergency response caused by
insufficient emergency facilities and resources. The proposed model represents a point of reference for the planning
and location of urban emergency facilities.

## 2. Multi-coverage Optimal Location Model Design


### 2.1 Problem description


Limited EMS resources face increasing demands as the risk of wide-scale and complex urban disasters increases.
Previous models have not considered the probability of failure of EMS facilities, in particular those housing
ambulances, nor have they taken into account possible limited access by EMS to vulnerable demand points. Hence,
two problems need to be addressed: first, the need for quick response times suggests that EMSs should be located
close to potential disaster points but far enough away to not be vulnerable; second, a high-risk area may require
simultaneous access to many EMSs. Based on these problems, in this study, we propose and formulate a disaster
scenario-based planning and optimal location model that considers multi-coverage of zones. The coverage is
dependent on the demand level of the demand points (high demand with high coverage requires more ambulances at
the same time). In our work, we specifically consider flooding; the location plan should result in improvements in
the efficiency of the response and reduce the risk to EMS of flash-flood disasters.
We present the objective of the proposed model and describe the problems encountered during the development of
the model. The objective of the model is to serve the largest number of people in a region with EMSs. Let J be the
set of potential emergency facilities, let I be the set of the demand points in the study area, and let F ($0<F<J$) be the
number of optimal facilities. We consider the risk of a disaster at the potential emergency points and the demand
points separately and arrange the station locations according to the coverage level and disaster risk level of each
demand point $i$. In simple terms, the model solves the following problems.
①  How do we calculate the coverage level $Q_i$ at each demand point $i$?
②  How do we evaluate the risk of disasters for each potential point j and demand point $i$?
③  What are the objectives and constraints for developing an optimized location model based on ① and ②?
④  How do we evaluate the applicability of the model?

**2.2 Assumptions**

To solve the above problems and simplify the model, we use the following assumptions:

① All potential points have the same probability of accepting EMS calls and their ability to serve all the demand points throughout the study area is not time-limited;

② Each emergency facility has the same service capacity and the same number of ambulances;

③ During a disaster, the closer the EMS is to the source of the disaster, the higher the probability is that the facility will be unable to respond;

④ During a disaster, the closer the EMS is to the source of the disaster, the greater the requirements placed upon it from any demand point.

**2.3 Mathematical model**

In accordance with the aforementioned description and assumptions, a multi-coverage optimal location model is developed. In the disaster scenario used for the model, it is assumed that each emergency facility has the same number of ambulances and quality of service and we must maximize the number of people it can serve within the specified time. In order to simplify the model and optimize the algorithm, we use the 0–1 integer programming method.

The model index sets are as follows.

$I$: set of demand points indexed by $i \in I = \{1, \dots, i, \dots, m\}$;

$J$: set of potential emergency medical facilities indexed by $j \in J = \{1, \dots, i, \dots, n\}$;

$t_{ij}$: time needed for an ambulance to travel from emergency medical facilities j to demand point i;

$X$: the number of demand points that can be covered by the service area of the emergency facilities within a specified time;

$T$: the limit of the emergency response time;

$F$: number of EMS facilities that need to be built;

$Q_i$: the coverage level of demand point i; meaning that point i should be covered by emergency facilities at least $Q_i$ times within a specified time;

$w_i$: the number of people represented by demand point $i$;

$m_i$: the disaster risk level of demand point $i$;

$p_j$: the resistance level to the disaster of potential point j;

$x_i$: binary value, equal to 1 if demand point $i$ is covered, otherwise, it is 0;

$y_j$: binary value, equal to 1 if an emergency medical facility is available, otherwise, it is 0;

$z_{ij}$: binary value, equal to 1 if demand point $i$ is covered by an eligible facility j, otherwise, it is 0.

The overall objective of the model is to rescue the maximum number of people in a specified time, as shown by the following equation:

$$\max (z) = \sum_{i=1}^{m} \sum_{j=1}^{n} (m_i\, w_i\, z_{ij}\, p_j) \tag{1}$$

To ensure the efficiency of rescue, the emergency response time must be minimized. To keep construction costs under control, the number of emergency facilities should be limited. Emergency facilities cannot be built in areas at risk of inundation and the coverage rate should be ensured within a specified time. Therefore, the following constraints are added to the multi-objective function:

$$\sum_{i=1}^{n} y_j = F \tag{2}$$

Constraint (2) indicates that F emergency facilities should be selected from the potential facilities for emergency





services;

$$\sum_{i=1}^{n} z_{ij} \left(1/p_j\right) \geq x_i \, Q_i \, (p_j \neq 0) \tag{3}$$

Constraint (3) ensures that the multiple coverage requirements of the demand points must be met under different
disaster scenarios and the resistance level $p_j$ to a disaster of potential point j cannot be 0;

$$t_{ij} \leq T \tag{4}$$

Constraint (4) ensures that the emergency response time cannot exceed T minutes;

$$x_i \geq X \tag{5}$$

Constraint (5) guarantees that $X$ demand points will be covered within at least T minutes;

$$z_{ij} \leq y_j (\forall i \in I; \forall j \in J) \tag{6}$$

Constraint (6) means that the service point can be serviced only when this facility is selected.

$$z_{ij} \in \{0,1\},$$
$$x_i \in \{0,1\}, \tag{7}$$
$$y_j \in \{0,1\}$$

Constraint (7) defines the domains of the decision variables.

**2.4 Coverage level analysis**

The model design indicates that the proposed model is based on a goal programming algorithm to optimize the
location of the EMS facilities based on the existing data and actual situation using by the coverage level $Q_i$ of each
demand point and the disaster risk level $m_i/p_j$ of the demand points/potential facilities. In this section, we propose
a new method to estimate the coverage level that depends on the demand level of demand point $i$.
Under normal conditions, the demand for EMS in one region is mainly related to population attributes such as age
distribution and population densities and areas of high population densities have a high probability of medical
emergencies. The surrounding conditions also affect demand, for example, traffic conditions and the presence of
regular medical services (such as hospitals). Therefore, in this study, we analyze the demand level based on these
related factors (labeled as evaluation indicators (A)) and we evaluate the probability of the demand point calling for
help within a predefined time. We then calculate the demand level of every point that is affected by these factors
considering the results in terms of the coverage level, i.e., how many times should point $i$ be covered by the service
area of the emergency facilities. Let A (A = $\{ A_1, \ A_2 \dots A_n\}$) be the set of indicators that may affect the coverage
level. In order to eliminate the influence of dimension and magnitude and improve the accuracy of the model, the
range normalization method is used to convert the original data into the range of [0,1]:

$$An_i = \frac{An_i - \min(An)}{\max(An) - \min(An)} \tag{8}$$

where $An_i$ represents the normalized index of the indicator set $A$.
These indicators determine the coverage level of demand and they have a certain weight:

$$Q_i = INT(\alpha A1_i + \beta A2_i + \cdots + \varepsilon An_i + 1) \tag{9}$$

where $\alpha, \beta \cdots \varepsilon$ represent the weights of the different indicators, i.e., their relative contributions to the estimated
demand. The coverage level $Q_i$ is then determined by increasing the integers; the results represent the number of
times this point needs to be covered by the emergency facilities.

**2.5 Disaster risk level analysis**



Events such as floods, earthquakes, and mudslides can adversely affect surrounding buildings and traffic and have
serious impacts on EMS. Not only is there is a high probability of casualties in the disaster source area, which creates
high demand for EMS but the disasters may cause road damage and traffic congestion, making rescue more difficult
than usual and delaying the emergency response. In order to achieve the model goal, we analyze the disaster risk
level of the demand points and potential emergency points and classify the disaster level according to the distance of
the emergency services from the source of the disaster. For a disaster risk level $m_i$ of demand point $i$, the closer the
point is to the location of the disaster, the higher the risk level and the probability of emergency calls for rescue are.
For the disaster risk level of the potential facility $j$, the closer the facility is to the disaster source, the more serious
the impact on the facility is, making its location unsuitable for an emergency facility. We express this indicator of the
alternative point as the disaster resistance capacity level $p_j$; therefore, the disaster resistance of the potential facilities
increases with their distance from the disaster source.

## 3. Case Study

For the case study, we choose Minhang District, Shanghai, China as the study area and apply the proposed location
model to the optimization of the EMS station distribution during the fluvial flooding hazards of Huangpu river based
on the data of the Shanghai Emergency Center.

### 3.1 Study area

Minhang district is located in Shanghai in China, covers an area of approximately 372.56 km$^2$, and is located near
the Huangpu River. There are 9 towns and 514 communities with about 253.4 million residents in the district. The
Huangpu River runs through the entire area and its river network consists of more than 200 rivers, making the study
area a high-risk area for fluvial flooding. In recent years, due to sea level rise and urban land subsidence, the risk of
storm surges and floods in the area surrounding to Huangpu River has increased (Yin, et al., 2013). In addition, part
of the Minhang district is in the center of Shanghai and has a complex road network and dense population. In addition,
long-term human activities have caused the natural river flow to decrease and the impervious surface areas in the
urban areas to increase, making the location highly vulnerable to pluvial floods and fluvial floods. There are currently
12 emergency stations in different blocks of this district and most stations are located downtown in the densely
populated areas (Fig. 1). Statistical data of the 2017 Shanghai Emergency Center indicates that the number of EMS
calls in 2017 exceeded 40,000 and the average emergency response time was about 15 minutes. When large-scale
flooding occurs, the emergency response efficiency is greatly affected due to the inundation of the road network.
Therefore, we considered a fluvial flood as a disaster scenario for applying the EMS location model.

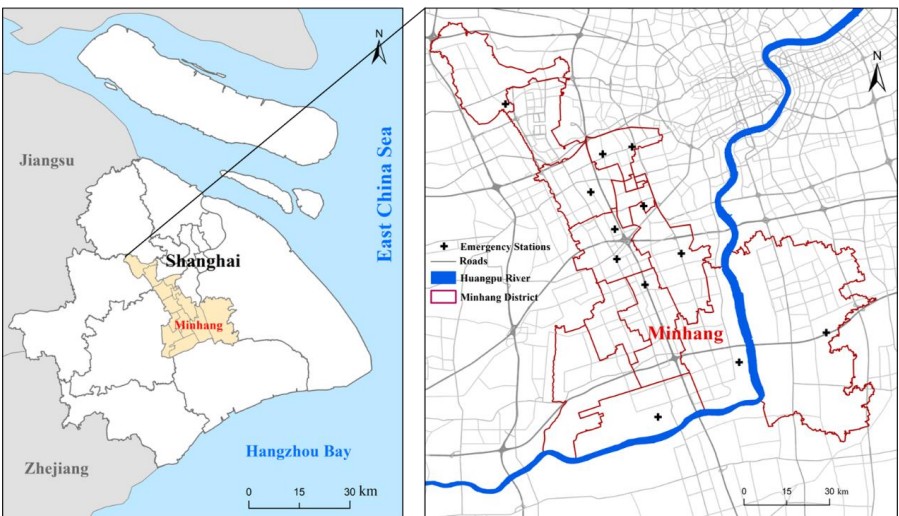

245            Fig. 1 .Location of the study area

**3.2 Flood impact analysis**

In order to assess the inundation area and depth following fluvial flooding disasters in the study area, we used a
1D/2D coupled flood inundation model, i.e., FloodMap(Yu and Lane, 2006a; Yu and Lane, 2006b), to simulate the
inundation scenarios of fluvial flooding in various return periods; this model combines the 1D solution of the Saint-
Venant equations of river flow with a 2D flood inundation model based on raster data to solve the inertial form of the
2D shallow water equations. The model is tightly coupled by considering the mass and momentum exchange between
the river flow and floodplain inundation and it is the now the mainstream numerical simulation model used for flood
scenarios (Yin and Yu et al., 2013; Yin and Yu et al., 2015). We use the FloodMap model to simulate the inundation
area and depth following fluvial flooding for various return periods (100-year and 1000-year) in the Huangpu River
Basin in the 2010s, 2030s, and 2050s (Fig. 2). The research data sources include the Shanghai 2013 Transportation
(Gaode) navigation GIS dataset, Shanghai public service facility data, a Shanghai 50-meter resolution digital
elevation model (DEM) and basic GIS data.

After obtaining the flood scenario simulation results, we used various (GIS) tools (e.g., the Spatial Analysis function
in combination with the Network Analysis function) to assess the impacts of flooding on the EMS response of the
existing emergency stations. We used the Shanghai Gaode GIS road network data and the 2017 EMS calls for help
data in the Minhang District obtained from the Shanghai Emergency Center. We used five levels for the road speed
limit based on the *People's Republic of China Technical Standard of Highway Engineering (JTG B01-2003)*. Our
assessment includes a network-based spatial analysis method using the road network data to derive areas that can be
reached from an EMS station within a certain timeframe. This method is widely used in route planning (e.g., via
Google Maps navigation) and considers speed limits, road hierarchy, one-way traffic, and other restrictions in the
road networks; this method is used by network analysis function in the ArcGIS10.2 software (New Service Area).
Given that the response time is the usual standard by which the efficiency of emergency rescue is assessed, we divided
the service area by using the ambulance travel time. In terms of the response time limit for ambulances, 8 min is
usually regarded as the standard for a medical emergency (Pons and Markovchick, 2002). However, the EMS calls
and rescue data from the Minhang District in Shanghai in 2017 indicated that the average medical emergency
response time was about 15 min, although the goal is to reduce this to 12 min by 2020. We have therefore used
response times of 8, 12, and 15 minutes to divide the EMS service area (Yin and Jing et al., 2019). In terms of
emergency management, When fluvial flood disasters occur, roads near rivers become inundated, leading to traffic
congestion and road closures, which affect ambulance response times, and the failure of some part of the transport
infrastructure would have the most serious effects on access to specific locations and overall EMS system
performance(Albano et al., 2014). Studies have shown that when road inundation reaches a depth of 30 cm, the roads
become impassable to vehicles (Yin and Yu et al., 2016; Green et al., 2017). We have, therefore, used an inundation
depth of 30 cm as the road closure restriction for vehicles; we used the same depth to define the area that cannot be
accessed by vehicles (the 'barrier area') in our GIS service area analysis. We used FloodMap to simulate flood
scenarios in 2010, 2030, and 2050 for two return periods (100-year and 1000-year). We then used the ArcGIS 10.2
network analysis toolbar to simulate the emergency facility service areas for the different scenarios (Fig. 2).

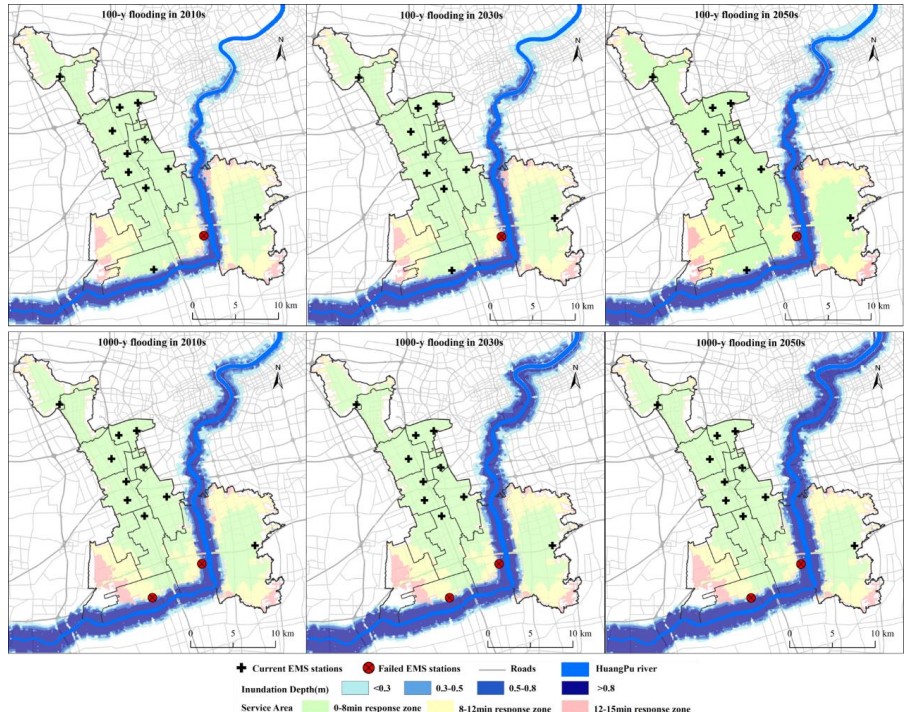


Fig. 2 Emergency station service areas in the Minhang District under different flood scenario simulations

Figure 2 shows that during a 100-y flooding occurs, one emergency station will lose capacity due to inundation,
whereas a 1000-y flooding will affect two stations, both of which are located near the middle and lower drainage
basin of the Huangpu River and serve a large population. If these two stations are incapacitated, it will greatly affect
the efficiency of medical emergency rescue in the surrounding areas. Figure 3 shows the impact on the area serviced
by each station for the different flood scenarios. The y-axis is the ratio of the service area before and after the disaster,
the lower the ratio, the greater the decrease is in the service area due to fluvial flooding. About half of the stations
are affected by the disaster and their service areas have decreased by more than 10%. The starting point for our
simulations is the distribution of the existing Minhang District emergency stations. We find that the existing EMS
distribution is inadequate for any of the flood scenarios used in our model. We, therefore, seek to optimize the location



of the emergency stations in conjunction with the flood scenarios to ensure that the emergency service facilities can
handle the disasters.

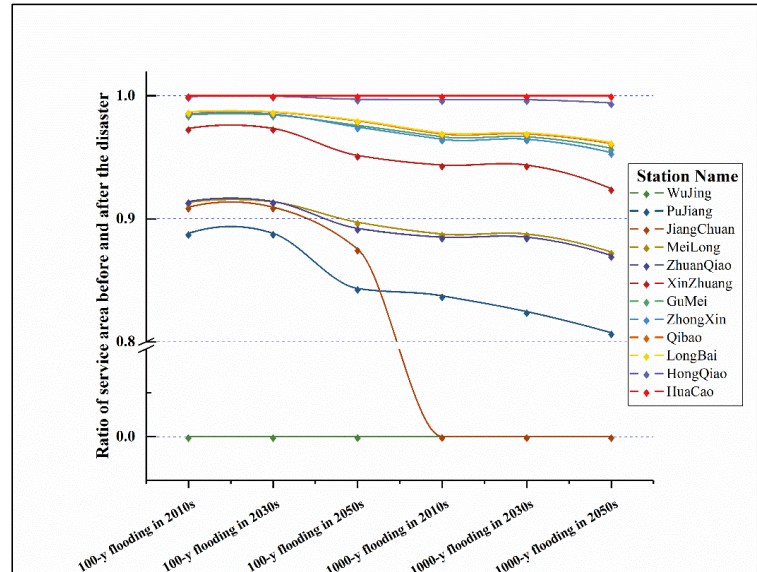


Fig. 3 Ratio of the service area of emergency stations before and after the disaster under different flood scenarios

**3.3 Model parameter calculation**

We calculated the two major model parameters (coverage level and disaster risk level) as proposed in Section 2 based
on the flooding scenario results described in Section 3.2 and used actual data for population, EMS calls for help, etc.
We first determined the demand points and number of potential emergency stations by dividing the study area into
units of representative blocks or grids. We used data on the location of the communities in the Minhang District to
determine the smallest block unit suitable for modeling demand (each community represents a demand unit). We
used the ArcGIS 10.2 software Geometry Calculation function to calculate the geometric center of each community
demand unit as a model demand point. To determine the location of potential EMS stations, we covered the whole
study area in a comprehensive and uniform manner. We divided the area into grids of a certain length and assumed
that every grid center point was a potential emergency station. Considering the actual conditions in the research area,
we divided the area into a grid with a cell size of 2 km * 2 km using the ArcGIS 10.2 fishnet analysis tool (create
fishnet). In addition, we added the original 12 emergency stations in the Minhang District to these potential stations
for comparison. There were 514 demand points and 106 potential stations in the study area (Fig. 4).

**3.3.1 Coverage level calculation**

The coverage level $Q_i$ of the demand points depends on the properties of each point. For example, the larger the
population, the more EMS stations are required and these should be located nearby. By considering the existing data
and the general conditions in the study area, we regarded the population and the historical EMS calls for help at each
demand point as the influencing factors $A_1$ and $A_2$, respectively of the demand coverage level (using Eq. (9)) and
used equal weights for the two factors ($\alpha = \beta = 0.5 * 10$). The resulting $Q_i$ is the coverage level, i.e., the number



of times that each demand point i should be covered by the emergency stations in the service area within a specified
time. The optimization objectives are to prevent delays in the emergency response caused by busy emergency stations
during a disaster and we constrained these objectives using $Q_i$. The results of the demand level calculation are shown
in Table 1.

**Tab.1 Demand point coverage level (sub-sample of the demand point data)**

| Point ID | Area | Latitude | Longitude | EMS calls | Population | Coverage level |
|---|---|---|---|---|---|---|
| 1 | 162411.9 | 31.1464 | 121.4014 | 74 | 5225 | 4 |
| 2 | 63454.85 | 31.14126 | 121.3974 | 44 | 3217 | 6 |
| 3 | 95601.05 | 31.1435 | 121.4021 | 59 | 3137 | 4 |
| 4 | 206827.6 | 31.13823 | 121.3785 | 89 | 5955 | 4 |
| 5 | 203574.8 | 31.13499 | 121.3794 | 150 | 6451 | 5 |
| 6 | 151097.8 | 31.13606 | 121.3842 | 173 | 4728 | 6 |
| 7 | 1463531 | 31.02184 | 121.4056 | 273 | 11332 | 2 |
| 8 | 631716.8 | 31.01324 | 121.4309 | 76 | 3317 | 1 |
| 9 | 3198358 | 31.055 | 121.4142 | 27 | 8736 | 1 |
| 10 | 130396.9 | 31.23009 | 121.3233 | 61 | 3970 | 4 |
| 11 | 129945.5 | 31.20503 | 121.2926 | 57 | 5082 | 4 |
| 12 | 307644.7 | 31.20974 | 121.2858 | 123 | 4113 | 2 |
| 13 | 254323 | 31.20926 | 121.291 | 71 | 3115 | 2 |
| 14 | 87982.62 | 31.20814 | 121.2776 | 51 | 4396 | 5 |
| 15 | 168857.8 | 31.24275 | 121.2677 | 37 | 4294 | 3 |
| 16 | 129736.7 | 31.23665 | 121.2693 | 69 | 3815 | 4 |
| 17 | 2101426 | 31.20482 | 121.2771 | 113 | 2801 | 1 |
| 18 | 3886865 | 30.99615 | 121.37 | 90 | 6481 | 1 |
| 19 | 217824.7 | 31.00576 | 121.3946 | 58 | 4066 | 2 |
| 20 | 302252.4 | 31.01126 | 121.3888 | 114 | 5911 | 3 |


### 3.3.2 Disaster risk level


The results of the disaster scenario analysis indicate that some existing emergency stations are themselves
highly vulnerable to fluvial flooding, which would delay or even prevent their EMS response. At this stage, we must
assess the disaster risk at all points before optimizing the locations of the emergency stations. We have considered
both the disaster risk level of the demand points and potential stations; a high risk level not only means that this
location is unsuitable for the location of EMS but it also indicates a high need for EMS.

We used the disaster risk analysis method proposed in Section 2.5. For the demand point risk level $m_i$, the disaster
risk level assessment of the potential stations and the demand points are classified by inundation depth. Point i in
the inundation area (depth of more than 30 cm) is regarded as completely inundated at the highest flooding risk level;
therefore, we use the area with the inundation depth greater than 30 cm as the center and create three 1 km wide
buffer zones $(m_i \in \{1,2,3\})$. The closer a point is to the inundation center, the higher the risk level of the demand
points (Fig. 5). In contrast, the risk level of the potential stations $p_j$ can be regarded as the resistance capacity to a
disaster; it increases with the distance to the inundated area. Therefore, we use the center of the inundation area with
a depth of greater than 30 cm and divide the disaster resistance level into four 1-km wide buffer zones ($p_j \in$
$\{0,1,2,3\}$). Hence   $p_j = 0$   means that the potential station j is completely inundated and cannot be used as an
emergency station.

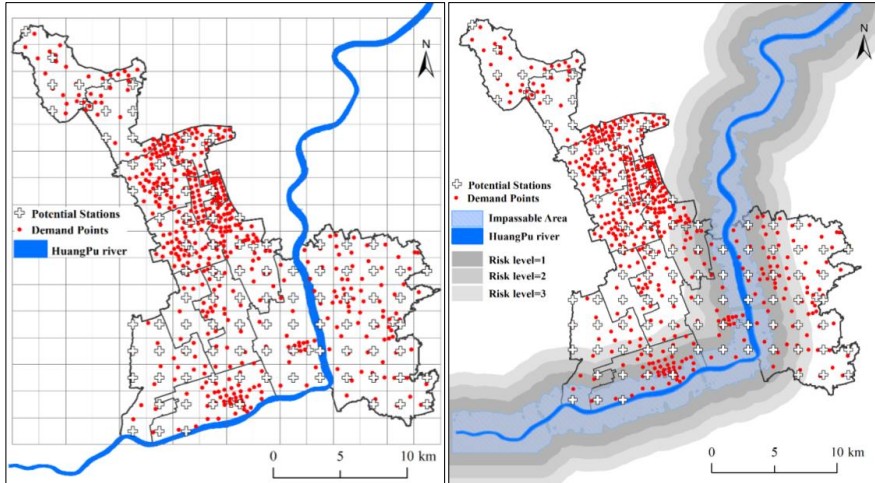

Fig. 4 Demand points and locations of potential stations   Fig. 5 Risk level for demand points and potential stations
**3.4 Results**
Here we present the results of the proposed multi-coverage optimal location model for the assignment of the Minhang
District emergency stations during fluvial flooding and discuss the performance of the optimization of the EMS
services and coverage level. In order to test our model, we run this model based on the worst-case scenario (1000-y
flooding in the 2050s). We have assumed that vehicles cannot travel through areas with inundation depths greater
than 30 cm. We utilized origin/destination (OD) matrix in the Network Analysis function of ArcGIS to calculate the
ambulance driving time   $t_{ij}$   from each potential station $j$ to each demand point $i$ during the disaster scenario. The
model also included the parameters for the construction of 12 stations  (F = 12)  to ensure that their service area
could cover at least 95% of the demand points within 8 min (X $\geq$ 514 $*$ 0.95, $t_{ij} \leq 8$). In simple terms, the objective
of this model was to determine the locations of emergency stations to rescue the largest number of people in 8 minutes.
We used the demand coverage level parameters and disaster risk level parameters obtained from the above-mentioned
analysis as inputs for the model and used Lingo10.0 software to solve the model. The computational results are given
in Fig. 6. The central urban area of the Minhang District is less affected by flooding than other areas; therefore, the
location of the EMS stations did not change significantly. However, in the region near the Huangpu River, the
optimized emergency stations are located farther away from the inundation area than the current stations, indicating
that the station at the optimized location will be less liable to flooding and more likely to remain operational than the
current stations.

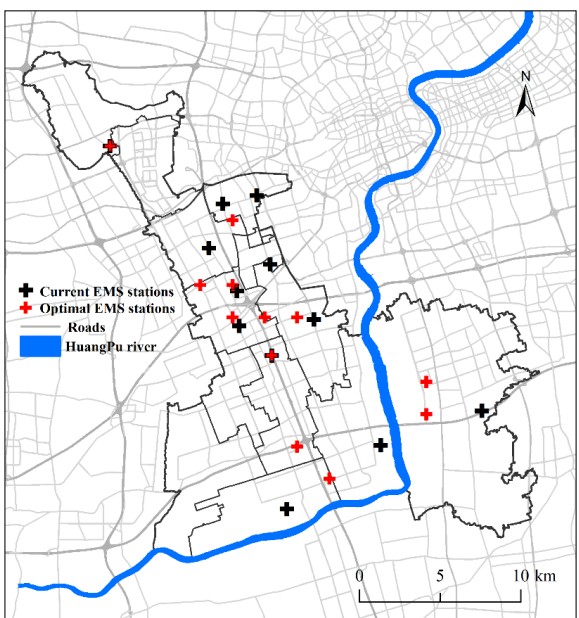


369                      Fig. 6 Computational results of the optimal location model


**3.4.1 Service capacity comparison**


In terms of emergency management, a service area is an intuitive measure for determining the service quality of emergency service facilities and usually reflects accessibility, i.e., the larger the service area, the larger the number of people who can be served by this station. In general, service areas and population are directly related to the transport infrastructure conditions around the emergency facilities, including road speed restrictions and road network density. During flooding, the transport infrastructure near the flooded area will be affected, which will change the travel time of the emergency vehicles, thus reducing the area of emergency service and accessibility of rescue. Therefore, in this context, we used the service area and population as parameters to evaluate the optimization performance of the model. Using the ArcGIS 10.2 Service Area Analysis tool, we divided the simulated emergency station service area into three response zones (8-, 12-, and 15-min journeys) under different scenarios; we then used the Spatial Join function to calculate the number of people in the service area. The total service area of the emergency stations for the different response times was calculated and the comparisons of the service capacity for the current stations and optimal stations are shown in Figure 7 and Figure 8 using the worst-case flooding scenario (1000-y fluvial flooding of the Huangpu River in the 2050s) and the no-flooding scenario.
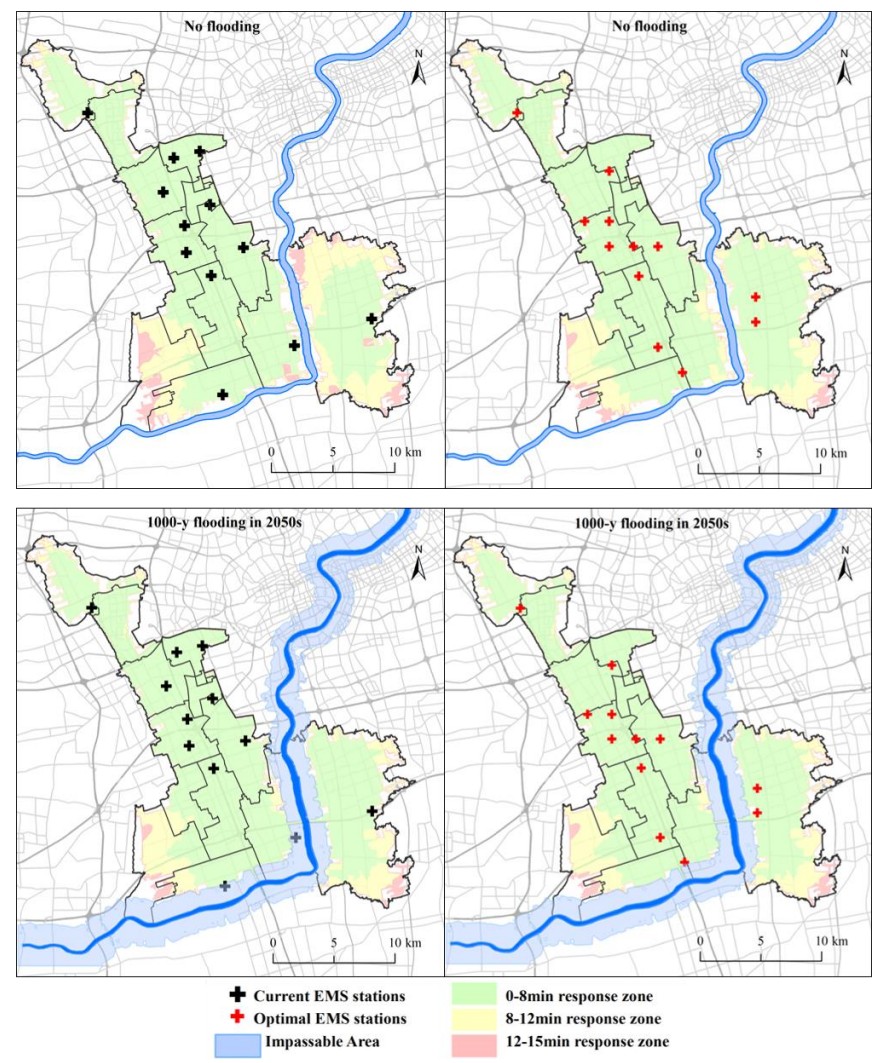


Fig. 7 Performance comparison of service areas in different scenarios



Fig. 8. Service capacity comparison


The percent coverage is expressed as a percentage of the total area and the total population; the results suggest that
the optimized locations of the emergency stations obtained by the model provided improvements in the service
capacity over that of the original stations in both the no-flooding and extreme flooding scenario based on the 8-min
emergency response time. In the no-flooding scenario, the coverage of the service area increased by about 5.5% and
for the worst-case flooding scenario, the increase was 8.4%. (Figure 8); the number of people with access to
emergency services increased by almost 250,000 (10% increase). These results indicate that the optimization model
increased the service capacity for almost all response times and the performance is best for the 8-min response time.

**3.4.2 Coverage level performance**

A combination of limited vehicle resources, vulnerable transport infrastructure, and high requirements of the demand
points during a disaster inevitably places emergency services under great pressure. If one demand point is covered
by only one emergency station, the limited number of ambulances would soon affect the provision of services for a
large number of demand points, thereby causing delays in rescue. Therefore, a region with high demand should be
covered by multiple emergency service areas that can operate simultaneously, especially for high-need demand points.
The proposed model focuses on multiple coverage levels of demand points and we used the average coverage level
for each demand point in a specific time as an important indicator to validate our model results. We combined the
service areas of all emergency stations and used the Spatial Join tool in ArcGIS 10.2 to calculate how many times
every demand point would be covered in 8, 12, and 15 minutes during the no-flooding and the worst-case flooding
scenarios. We then compared the average values (Figure 9).

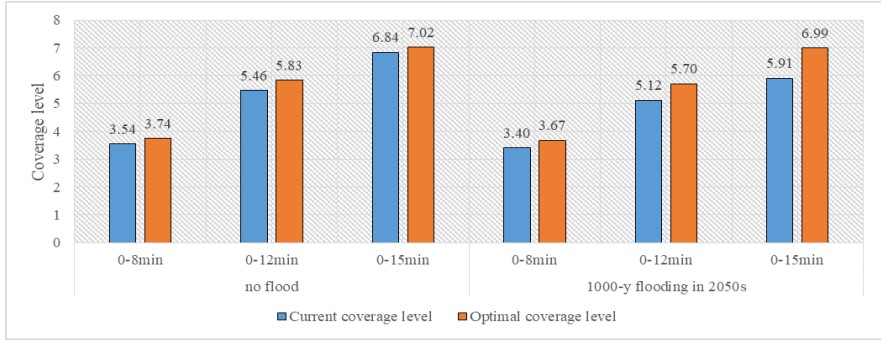


411                      Fig. 9 Comparisons of the average coverage level

Figure 9 shows that the average coverage level improved after optimization in both scenarios. Specifically, the
average coverage level for the no-flooding scenario is slightly higher (about 10%). The improvement in the coverage
level for the no-flooding scenario was greatest for the 12-minute response time, i.e., an increase of 6.8%. For the
worst-case flooding scenario (1000-y fluvial flooding of the Huangpu River in the 2050s), the improvements were
more significant: the coverage of the 15-minute response time increased by more than one (18.4%), indicating that,
on the average, each demand point can be served by one additional EMS stations within 15 min. These results indicate
that using model optimization for locating emergency stations greatly improved the response time of emergency
services at the demand points, even in an extreme flood disaster scenario, thereby providing strong disaster resistance.
The comparison also shows that stations whose locations are determined using the proposed method will have a
greater capacity to meet the requirements of the demand points. This reduces the occurrence of "failures" and
"insufficiency" of emergency stations during disasters, thereby shortening emergency response times and reducing



the loss of life and property.

## 4. Conclusions

This study focused on the optimization of the EMS station locations to ensure efficient emergency medical response
in fluvial flood disaster scenarios and the prevention of accidents due to emergency response delays and failure of
stations. After analyzing the existing location models, we discussed the reasons for using multi-coverage plans to
improve disaster emergency resistance instead of traditional location models. In addition, since there are various
disaster scenarios, we also considered the different damage levels in various areas using disaster scenario simulations.
The proposed model is an objective programming model with the goal to serve the largest number of people in a
specified time during a disaster. For the case study, we investigated the Minhang District in Shanghai, China and
conducted computational experiments based on real-world data from the Shanghai Emergency Center. We used the
service area and the average coverage level as parameters to evaluate the model performance. The model results
showed that the optimized EMS locations had a wider service range for 8-min response time and a larger number of
people were served; the coverage level was also improved. The coverage level of some of the existing stations
changed greatly after the disaster whereas the optimized location results showed that the service level before and
after the disaster was almost the same. Both parameters indicated that the proposed multi-coverage location
optimization model is well suited to model the emergency response to flood disasters and to conduct site selection of
urban emergency facilities.

The model also has some aspects that could to be improved in order to arrive at more robust solutions. Firstly, in our
case study, we did not have a quantitative assessment of the disasters risk level on emergency response, we evaluated
the disaster risk level only by the buffer distance to disaster source area, which is subjective. Secondly, as we only
analyzed in theory, our model did not consider whether the terrain or other basic conditions were suitable for the
EMS facilities, and the future studies will take the real terrain and construction cost of each potential point into full
account.

Lastly, the location of urban emergency service facilities has always been the focus of urban planning. Location
selection should consider a variety of factors and the ability to respond to disasters is also a key factor to consider. In
this study, we used a fluvial flooding disaster as an example to analyze the impact of disasters and to evaluate the
model. However, the risks faced by cities are not only fluvial floods but also other major events such as earthquakes,
mudslides, and pluvial floods. What's more, the evacuation plan of the population exposed to these hazards should
be considered (Alaeddine, 2015). Future research should comprehensively consider a variety of these hazards,
conduct risk assessments of the study area quantitatively, and select the location of urban emergency facilities
according to different geographical conditions to improve the efficiency of emergency response.

**Acknowledgments**: This work was supported by the National Key Research and Development Program of China
(Grant no: 2017YFE0100700), the National Natural Science Foundation of China (Grant no: 41871164), and the
Humanities and Social Science Project of Education Ministry of China (Grant no: 17YJAZH111).







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
