# Peer review of "Multi-coverage Optimal Location Model for Emergency Medical Services (EMS) facilities under various disaster scenarios:A case study of urban fluvial floods in the Minhang District of Shanghai, China"

_Natural Hazards and Earth System Sciences, 2019_

## Referee Comment (RC1) · Kamal Serrhini (Referee) · 26 Aug 2019

A. Questions related to the contain of the paper

Yuhan Yang and Jie Yin, 2019, Multi-coverage Optimal Location Model for Emergency Medical Services (EMS) facilities under various disaster scenarios: A case study of urban fluvial floods in the Minhang District of Shanghai, China", NHESS.

The paper focuses on a *"multi-coverage optimal location model for Emergency Medical Services (EMS) facilities based on the results of disaster impact simulation and prediction"* (p. 1) ». *"The main purpose of our location model is to reduce the probability of delays in the emergency response caused by insufficient emergency facilities and resources"* (Lines 94-95). "… **the objective of this model was to determine the locations of emergency stations to rescue the largest number of people in 8 minutes**." (Lines 359-360).

This article focuses on optimizing the location of 12 EMS (China, Shanghai, Minhang District) by considering 514 demand points and 106 potential EMS points. For this, the authors combine 3 complementary aspects/steeps:

- Approach of the optimization of the EMS location (with Lingo 10.0 software)
- A flood simulation approach (1D and 2D models). The software used is not indicated? However, two references are mentioned (Yu and Lane 2006a and 2006b).
- And geomatic processing (GIS) with ArcGIS 10.2 (especially Network Analyst module).
* * *
Line 86: "... *here we describe a* ==novel== *approach for the optimization of EMS*"

And line 189: "*we propose a* ==new== *method to estimate the level of demand*"

What is/are the innovative aspect-s of the paper?

- The implementation of a treatment chain including the development of flood scenarios (100 and 1000 years return periods).
- An interesting aspect is the "*Coverage level analysis*" (2.4 section).

The aspect of "*Disaster risk level*" analysis (2.5 section) simply depends on the proximity of the flood hazard and EMS (Euclidean distance? See line 215).
* * *
Assumption number ② (line 131): in general, the EMS capacity and the number of ambulances are variables by EMS (for each EMS). These aspects would constitute future developments (perspective) for improving the model proposed (see below).
* * *
Line 163 : "*To ensure the efficiency of rescue, the emergency response time must be minimized*": for each ambulance (each rescue) or for all ambulances (all calls/rescues)?

The aim is to minimize the time (OD) between i (demand point) and j (facility or EMS point) OR to minimize the total time of all trips/journeys between all demand points (i) and EMS (j)?

If only the travel time of each ambulance must be optimized (independently of the trips of the others ambulances), the corresponding OD distance must indeed be minimized.

[ As you know, during a crisis period, the emergency vehicles can travel relatively faster compared to regulatory speed limits according to the traffic jam (the ambulance use there sirens). On the other hand, the urban density tends to homogenize and to reduce the statutory road speeds of 30, 50, 70 to 90 km/h between the city center and the countryside areas. ]

However, calls usually come to a call center, which distributes them according to different aspects, such as the availability of ambulances, the remaining capacities of the nearest EMS of the site, and so on. But it seems that the paper is not in this configuration (sorry, I am not familiar with the Chinese rescue system). Please, see the comment above related to the assumption ②.

The minimization of the total time (sum) of all the journeys between the demand points (i) and EMS (j) induce the problem of competition between calls and therefore peoples to be rescued: first come, first served (beds in a hospital / EMS).

Hence, the combinatorial aspect (complex problem, NP ...) of such assignment depends on: the number of peoples to be rescued per calls / demand points (i); the capacity of each EMS (j) and available ambulances / vehicles and their own / specific capacities (equipment).

[ Other aspect: not all EMS will be able to welcome some people suffering from a particular pathology / disease / trouble (asthma or heart attack due to anxiety and fear of flooding). For such cases, the ambulance must be equipped with specific equipment/material plus qualified stretcher (nurses). This neglected aspect can be indicate as future perspective. ]
* * *
Line 167 :

$\sum (i=1 \text{ to } n) \text{ of } y_j = F$

Or

$\sum (j=1 \text{ to } n) \text{ of } y_j = F$ ?
* * *
Line 255: "…*in the Huangpu River Basin in the 2010s, 2030s, and 2050s* (Fig. 2)"

Does this mean that the flood simulation model takes into account aspects such as precipitation trends, urban sprawl and / or population change in 2010, 2030 and 2050 (in a context of climate change?). I imagine that all these aspects are considered in the cited references of Yu and Lane 2006a and 2006b (Line 249).
* * *
Line 263 : "*We used five levels for the road speed limit*"

Remember that ambulances and rescue services (fire brigade) in general are allowed to exceed speed limits during an intervention. For low speed road sections (30 km/h for example), we could increase this speed in the model... (under ArcGIS, it is quite possible / easy to change the speed of a category / class of road sections with VB or Python script).
* * *
Section 3.3:

I understand that the potential stations (106) are the centroids of squared meshes (fishnet) of the applied grid (2 km * 2 km) on the site.

On the other hand, the method/process of designing the 514 demand points is less clear (red dots, Fig. 4 - 5, page 11). Shanghai → Minhang district → Community unit (demand unit = smallest block unit)?

Does each red dot correspond to a block of buildings (group of contiguous buildings)? If yes, how to know the population by block/group of buildings (in China)? Do you know the population per building? (see section 3 Grouping of Buildings, *Alaeddine et al*., 2015, pages 689-690).

About the applied grid of **2 km * 2 km**: can be discussed in the section of results / discussion / conclusion of the article. Indeed, what is the impact of such division/regular zoning on the method and results obtained? Can we develop / imagine a **multi-scale** division with variable squared meshes taking into account the distribution density of the population (spatial distribution of red dots)?

Please see the example of map 1, page 17 of the paper:

https://www.researchgate.net/publication/26431851_Integration_quantitative_du_paysage_lors_de_la_determination_de_traces_d%27un_amenagement_lineaire
* * *
Line 327: Tab 1. No need to display/show the coordinates of points 1, 2, 3, etc. (latitude and longitude values). However, it misses the values of: min (A1), min (A2), max (A1) and max (A2) (equation 8) to allow the reader (who wishes to do it) to calculate/verify $Q_i$?
* * *
Point ID 1 (Tab.1) generated 74 EMS calls and the coverage level calculated $Q_i$ is equal 4.

Where $Q_i$ is : "*the number of times that each demand point i should be covered by the emergency stations in the service area within a specified time*" (lines 322-323).

So, do you know the number of real trips (statistical data of 2017)[1] done by ambulances between EMS and Point ID 1 (or at least the ratio between calls and trips)? If possible to
* * *
[1] Line 240, page 6: Statistical data of the 2017 Shanghai Emergency Center indicates that the number of EMS calls in 2017 exceeded 40,000 and the average emergency response time was about 15 minutes.

compare the distribution of $Q_i$ (calculated values) with the values observed on the site in the recent past (a way to appreciate/validate the values obtained/calculated of $Q_i$).
* * *
Line 340, Section 3.3.2: which flood scenario is considered in Fig. 5?

The 3 buffers of **1 km each** used to characterize the indicator "*Disaster Risk Level*" are more relevant (pertinent) especially if it is the flood scenario of 100-y (rather than 1000-y). In this case, the spatial discretization (by the 3 buffers) will be interesting to take into account the variability (uncertainty) of the flood extension between the simulated scenario and the observed one (flooding closer to 150 or 200-y ... than 100-y).

A flood of 1000-y, may already be considered as an extreme event (I am not familiar with the site studied). To have water beyond the flooded area of 30 cm (Fig.5), it would take a more extreme flood event (1500-y ...). Is this possible in the context of the study site (climate change)? In the past, has there been a higher (historical) flood than the 1000-y scenario?

Here is a proposal for the flood of 1000-y:

- $p_j = 0$ if water height is > 30 cm (EMS is completely inundated)
- $p_j = 1$ if water height is <= 30 cm
- $p_j = 2$ if water height is = 0 cm (EMS is not inundated)?

The method shown in Fig.5 seems more suitable (pertinent) for the 100-y flooding scenario.
* * *
Line 355: The calculation of the OD matrix with the ArcGIS NetworkAnalyst extension does not take into account the traffic jam during the crisis? (see Line 192)

The implementation of a transport model (traffic model) can be considered as a significant improvement (perspective) of the proposed model especially during crisis (flash flooding).
* * *
Line 374 : "…*i.e., the larger the service area, the larger the number of people who can be served by this station*": this statement (affirmation) is not always true.

In the center of the cities (metropolitan area), the service area of an EMS (or "shelters" in general) can be relatively small but with a large / high number of people to take care of. It therefore depends on the urban location and the capacity of each EMS.

Please see Figure 1, page 3 of the paper: « Optimisation combinatoire de l'affectation interne de la population de Nice aux centres d'accueil en cas de séisme », 2017. https://www.researchgate.net/publication/322040849_Optimisation_combinatoire_de_l'affectation_interne_de_la_population_de_Nice_aux_centres_d'accueil_en_cas_de_seisme/

Or see: https://hal.archives-ouvertes.fr/SAGEO2017/hal-01650670

In the present case of this article, the service area is calculate only according to the ambulance travel time ($t_{ij}$). It is an area of the same accessibility (isochronic or isodistance area). The higher the $t_{ij}$ (8, 12, 15 min), the larger the area ("and the larger the population") to be cared for may be important.

Hence, the importance to consider (in the future) the capacity of EMS (perspective).
* * *
Finally, what about the indirect impact of flooding (indirect vulnerability)?

Some doctors, drivers of ambulances, nurses, fire brigade agents, etc. living outside the flash flooded area will probably not be able to join their job location/office/building (due to the 'barrier area' effect, Line 281)…

B. Questions related to the form of the article ("technical corrections")

Questions ①, ②, ③ and ④ are not repeated / recalled ... in the section dedicated to the results. Example: Line 318: "The coverage level $Q_i$ of the demand points (question 1, line 119, page 3) ..."

There is a risk of confusion between Questions ①, ②, ③ and ④ (page 3) and Assumptions ①, ②, ③ and ④ (page 4). Use: Q1, Q2, Q3 and Q4 // A1, A2, A3 and A4 or other solution.
* * *
Line 188 : "… *the disaster risk level mi/pj of the demand points/potential facilities*"

mean:

*… the disaster risk levels of the demand points ($m_i$) and potential facilities ($p_j$) separately*? (see Line 116 : "*We consider the risk of a disaster at the potential emergency points and the demand points separately*")

Could you please precise / reformulate this line.
* * *
Line 242: for the reader unfamiliar with the study site and in terms of urban vulnerability, the selected flash flood scenario impact how many km of roads and how many peoples (buildings), the duration (number of hours, days…) of flooding…? If possible to provide more relevant /precise ideas about the flood impacts on the site.
* * *
Fig 1: I suggest considering another color than RED one to represent district boundaries.

In general, the legend of maps (Fig 1, 2 ...) are too small (not visible).
* * *
Concerning the two selected flood scenarios (100 and 1000-years), what is the major historical flood that has been observed on the site? It would be interesting to consider the major historical hazard? (section 3.2, line 255)
* * *
Line 274: In terms of emergency management, when  fluvial flood disasters occur, roads near rivers become inundated, leading to traffic…

Line 287: "*Figure 2 shows that during a 100-y flooding occurs, one emergency station (*==Name = ?==*, see Fig. 3) will lose capacity due to inundation*"

Line 288 : "*whereas a 1000-y flooding will affect two stations (*==Names = ?==*)*"
* * *
Line 290: "*Figure 3 shows the impact on the area serviced by each station for the different flood scenarios.*"

For the Fig 3, is it possible de precise ("again") the duration considered finally to compute (with ArcGIS, Network Analyst, Service Area Analysis) the best (shorted) trip of the ambulance 8, 12 or 15 min?
* * *
Line 298: it is difficult to make visually and easily the link between the coloured curves and the legend (the name of each EMS).

At least, the order of the name of the stations (12 EMS) in the legend must be the same than the order of the curves to improve the reading of this Fig.
* * *
Line 322: why the alpha and beta weights are the same (equal)?
* * *
Line 411: Fig. 9 Comparisons of the average coverage level

Figure 9 shows "*coverage level*" in REAL values (3.54, 3.74 etc.) and not in **INTEGER** values? (See equation 9, page 5)
* * *
Finally, I propose to the authors (if possible) to design a logi-gram related to the developed methodology and results. Please, see example of the Figure 2, page 689, *Alaeddine et al*., 2015.

Dr. K. Serrhini

---

## Author Comment (AC1) · 19 Sep 2019

We sincerely thank Reviewer #1 for his support for publication

**A. Questions related to the contain of the paper**

**1. A flood simulation approach (1D and 2D models). The software used is not indicated?**

Thanks for noting this. It's a very important question, the flood inundation model we used is called

FloodMap, this model is an established diffusion-based flood inundation model (FloodMap, Yu

2005; Yu and Lane 2006a, b, 2011)

We will add the introduction about this model in more detail as below:

.

"we used a 1D/2D coupled flood inundation model named FloodMap (Yu and Lane, 2006a; Yu and

Lane, 2006b), to simulate the inundation scenarios of fluvial flooding in various return periods; this model combines the 1D solution of the Saint-Venant equations of river flow with a 2D flood inundation model based on raster data to solve the inertial form of the 2D shallow water equations.

The model is tightly coupled by considering the mass and momentum exchange between the river flow and floodplain inundation and it is employed to simulate the flood process and extract flood potential maps. It has been applied in a number of different environments and now Floodmap is the mainstream numerical simulation model used for flood scenarios (Yin and Yu et al., 2013; Yin and

Yu et al., 2015). We use the FloodMap model to simulate the inundation area and depth following fluvial flooding for various return periods (100-year and 1000-year) in the Huangpu River Basin in the 2010s, 2030s, and 2050s."

**2. What is/are the innovative aspect-s of the paper?**

Thanks for your summary, this research mainly proposed multi-coverage location optimization model well suited to model the emergency response to flood disasters and to conduct site selection of urban emergency facilities.

The innovative aspects are

•    Improving the emergency service capacity from the aspect of service population and the coverage level(how often the demand point needs to be covered by emergency facilities) during disasters

•    The implementation of a treatment chain including the development of flood scenarios (100

and 1000 years return periods).

•    An interesting aspect is the "Coverage level analysis"

**3. The aspect of "Disaster risk level" analysis (2.5 section) simply depends on the proximity of**

**the flood hazard and EMS (Euclidean distance? See line 215).**

Thanks for noting this. In our case study, we used ArcGIS 10.2 buffer tool to determine the Disaster risk level by Euclidean distance. Because the impact of fluvial flood hazards on emergency response is directly related to inundated areas, unlike other disasters such as earthquake and mudslide, flooding does not destroy buildings on a large scale (the disaster risk will be related to whether the buildings are strong or not), so in our case study, we analyzed the disaster risk level of the demand points and potential emergency points and classify the disaster level according to the distance of the emergency services from the source of the disaster. In future research, we will try to develop a quantitative assessment of the disasters risk level on emergency response, is considered to be reasonable.

We have added this discussion in Conclusion as below:

"The model also has some aspects that could to be improved in order to arrive at more robust solutions. Firstly, in our case study, we did not have a quantitative assessment of the disasters risk level on emergency response, we evaluated the disaster risk level only by the buffer distance to disaster source area, which is subjective……. The future studies will consider disaster risk factors such as the vulnerability of buildings comprehensively, evaluate the level of disaster risk quantitatively, and take the real terrain and construction cost of each potential point into full account."

**4. Line 163: "To ensure the efficiency of rescue, the emergency response time must be**

**minimized": for each ambulance (each rescue) or for all ambulances (all calls/rescues)?(need**

**more detail )**

Thanks for noting this. Yes, in line 146 we defined that parameter $t_{ij}$ was the time needed for an ambulance to travel from emergency medical facilities j to demand point i. We use $t_{ij}$ to constraint that the emergency response time cannot exceed T minutes in model ($t_{ij}$<T), which met Chinese emergency response time limit. In time limit, how to serve the largest number people is the objective of our model.

The sentence has been removed to Line 173 and changed as below:

"Constraint (4) ensures that the emergency response time cannot exceed T minutes to ensure the efficiency of rescue;"

**5. However, calls usually come to a call center, which distributes them according to different**

**aspects, such as the availability of ambulances, the remaining capacities of the nearest EMS**

**of the site, and so on. But it seems that the paper is not in this configuration (sorry, I am not**

**familiar with the Chinese rescue system). Please, see the comment above related to the**

**assumption ②.**

Thanks for your comments. Yes, in normal cases, ambulances distributed according by distance or other aspects. We also analyzed almost 40000 records of EMS calls of study in 2017(Figure 1). The results show that demand points can be served by multiple EMS stations. Therefore, the assumption

②(During a disaster, Each emergency facility has the same service capacity and the same number of ambulances;) is based on one single historical data and this is considered to be reasonable especially during disasters.

[Figure]

Fig.1 the EMS calls records of 13 counties (Minhang district) in 2017

**6. Line 167 :** $\sum_{i=1}^{n} y_j = F$ **Or** $\sum_{j=1}^{n} y_j = F$

Thanks for noting this. It has been corrected.

**7. Line 255: "…in the Huangpu River Basin in the 2010s, 2030s, and 2050s (Fig. 2)"   Does this mean that the flood simulation model takes into account aspects such as precipitation trends, urban sprawl and / or population change in 2010, 2030 and 2050 (in a context of climate change?). I imagine that all these aspects are considered in the cited references of Yu and Lane 2006a and 2006b (Line 249)**

Thanks for noting this. Yes, the model(Floodmap) we used is a mature flood inundation model, and in our case study, the flooding inundation simulation results was took reference by Yin et al (2013) research, it considered sea level rise and land subsidence on storm tides induced flooding of the Huangpu River(Figure 2).

[Figure]

Fig.2 multiple scenario of Huangpu river flooding

**8. Line 263 : "We used five levels for the road speed limit"**

**Remember that ambulances and rescue services (fire brigade) in general are allowed to exceed speed limits during an intervention. For low speed road sections (30 km/h for example), we could increase this speed in the model... (under ArcGIS, it is quite possible / easy to change the speed of a category / class of road sections with VB or Python script).**

Thanks for noting this. Yes, in general, ambulance are allow to exceed speed limit. However, in fact, road conditions included height and weight always constraint road speeds, and not all roads have emergency vehicle lanes that's why ambulances are not so easy to exceed speed limits. Furthermore, there are too much uncertainties associated with how human behavior and patterns of congestion may differ under flood conditions. Therefore, the speed of the road is difficult to define accurately.

**9. On the other hand, the method/process of designing the 514 demand points is less clear (red -dots, Fig. 4 - 5, page 11). Shanghai Minhang district Community unit (demand unit = smallest block unit)?**

Thanks for noting this. Yes, in order to verify the model applicability, we set each community unit as the smallest unit, because in China, the EMS services always allocated by blocks or communities, while the same communities have same attributes, so there is reason to take community unit as the
smallest unit. Another reason is we have Shanghai Communities' population and other detailed data,
what make study more precisely. Because of the communities are small, it is scientific to take the
central point as the rescue unit. Of course, it would be better if we had the building data, but the
efficiency of model running would also increase significantly.
**10. About the applied grid of 2 km * 2 km: can be discussed in the section of results / discussion/**
**conclusion of the article. Indeed, what is the impact of such division/regular zoning on the**
**method and results obtained? Can we develop / imagine a multi-scale division with variable**
**squared meshes taking into account the distribution density of the population (spatial**
**distribution of red dots)?**
Thanks for your suggestions. We have adde the discussion about the impact of such division/regular zoning on the method in conclusion:

"Lastly, the location of urban emergency service facilities has always been the focus of urban
planning. Location selection should consider a variety of factors and the ability to respond to
disasters is also a key factor to consider, while in this paper, we divided the area into grids with a
cell size of 2 km * 2 km and assumed that every grid center point was a potential emergency station,
The division of grid will affect the efficiency of model running efficiency and the accuracy of results.
The smaller the scale, the higher the accuracy, but the greater the model running pressure. Therefore,
in the future research, we will consider multi-scale division with variable squared meshes taking
into account the distribution density of the population or other factors. "
**11. Line 327: Tab 1. No need to display/show the coordinates of points 1, 2, 3, etc. (latitude and**
**longitude values). However, it misses the values of: min (A1), min (A2), max (A1) and max (A2)**
**(equation 8) to allow the reader (who wishes to do it) to calculate/verify Qi?**
Thanks for your comments, we have altered the table in revised paper (shown in Table 1)

**Tab.1 Demand point coverage level (sub-sample of the demand point data)**

| Point ID | Area(km²) | Population | EMS calls | Population density(A1) | EMS calls density(A2) | Coverage level |
|---|---|---|---|---|---|---|
| 1 | 0.1624119 | 5225 | 74 | 32,171.28 | 455.6315 | 4 |
| 2 | 0.06345485 | 3217 | 44 | 50,697.46 | 693.4064 | 6 |
| 3 | 0.09560105 | 3137 | 59 | 32,813.45 | 617.148 | 4 |
| 4 | 0.2068276 | 5955 | 89 | 28,792.10 | 430.3101 | 4 |
| 5 | 0.2035748 | 6451 | 150 | 31,688.60 | 736.8299 | 5 |
| 6 | 0.1510978 | 4728 | 173 | 31,290.99 | 1,144.95 | 6 |
| 7 | 1.463531 | 11332 | 273 | 7,742.92 | 186.5352 | 2 |
| 8 | 0.6317168 | 3317 | 76 | 5,250.77 | 120.3071 | 1 |
| 9 | 3.198358 | 8736 | 27 | 2,731.40 | 8.441831 | 1 |

| | | | | | | |
|---|---|---|---|---|---|---|
| 10 | 0.1303969 | 3970 | 61 | 30,445.52 | 467.8027 | 4 |
| 11 | 0.1299455 | 5082 | 57 | 39,108.70 | 438.6454 | 4 |
| 12 | 0.3076447 | 4113 | 123 | 13,369.32 | 399.8118 | 2 |
| 13 | 0.254323 | 3115 | 71 | 12,248.21 | 279.1726 | 2 |
| 14 | 0.08798262 | 4396 | 51 | 49,964.41 | 579.6599 | 5 |
| 15 | 0.1688578 | 4294 | 37 | 25,429.68 | 219.1193 | 3 |
| 16 | 0.1297367 | 3815 | 69 | 29,405.72 | 531.8465 | 4 |
| 17 | 2.101426 | 2801 | 113 | 1,332.90 | 53.773 | 1 |
| 18 | 3.886865 | 6481 | 90 | 1,667.41 | 23.15491 | 1 |
| 19 | 0.2178247 | 4066 | 58 | 18,666.38 | 266.2691 | 2 |
| 20 | 0.3022524 | 5911 | 114 | 19,556.50 | 377.1681686 | 3 |
| … | … | … | … | … | … | … |
| Max | 10978496.3425 | 25419 | 608 | 76608.25 | 1870.493324 | 8 |
| Min | 20271.96894 | 86 | 0 | 25.7722 | 0 | 1 |

**12. So, do you know the number of real trips (statistical data of 2017)1 done by ambulances between EMS and Point ID 1 (or at least the ratio between calls and trips)? If possible to compare the distribution of Qi (calculated values) with the values observed on the site in the recent past (a way to appreciate/validate the values obtained/calculated of Qi).**

Thanks for your comments. Sorry, we don't have the real trip of ambulances, but we analyzed the EMS calls records of 13 counties (Minhang district) in 2017(Figure 1), the vertical axis is the source county of the demand points and horizontal axis is the number of EMS calls. The results showed that each demand point can be served by multiple EMS stations, however, in normal cases the ambulance are always allocated by distance especially in a short time. For example, the demand points of some counties, such as Xinzhuang county, can be served by multiple EMS service stations (with high coverage level Qi), while some counties such as Jiangchuan county can only be served by Jiangchuan EMS station in most cases (with low coverage level Qi), which means that if the station is destroyed disasters (eg.1000-y fluvial flooding in 2050s), the emergency response time of Jiangchuan county will be greatly delayed.

We also compared how many times every demand point would be covered in 8, 12, and 15 minutes during the no-flooding and the worst-case flooding scenarios (Figure 3). The percent coverage is expressed as percentage of demand points for different coverage levels. The results were interesting, for example, in 8-min response time or 12-min response time, the model greatly improved the coverage level of interval 5~8, what's more, we also found that the optimized coverage level almost same during the no-flooding or the worst-case flooding scenarios, what means extreme fluvial flooding have little impact on EMS emergency response.

We will supplement the results in revised paper

[Figure]

                 Fig 3 Comparisons of the coverage level

**173 13. Line 340, Section 3.3.2: which flood scenario is considered in Fig. 5?**

**174 The 3 buffers of 1 km each used to characterize the indicator "Disaster Risk Level" are more**

**175 relevant (pertinent) especially if it is the flood scenario of 100-y (rather than 1000-y). In this**

**176 case, the spatial discretization (by the 3 buffers) will be interesting to take into account the**

**177 variability (uncertainty) of the flood extension between the simulated scenario and the**

**178 observed one (flooding closer to 150 or 200-y ... than 100-y).**

**179 A flood of 1000-y, may already be considered as an extreme event (I am not familiar with the**

**180 site studied). To have water beyond the flooded area of 30 cm (Fig.5), it would take a more**

**181 extreme flood event (1500-y ...). Is this possible in the context of the study site (climate change)?**

**182 In the past, has there been a higher (historical) flood than the 1000-y scenario?**

**183 Here is a proposal for the flood of 1000-y:**

**184 •pj = 0 if water height is > 30 cm (EMS is completely inundated)**

**185 • pj = 1 if water height is <= 30 cm**

**186 • pj = 2 if water height is = 0 cm (EMS is not inundated)?**

**187 The method shown in Fig.5 seems more suitable (pertinent) for the 100-y flooding scenario.**

Thanks for noting this. They are very important comments.

(1)We used the 1000-y fluvial floods of Huangpu River as the extreme flood scenario because in

Huangpu River, to protect against flooding, flood walls have been built since the 1950s. This has since been reinforced and upgraded, resulting in most of the study area along the Huangpu River being protected by 511 km flood walls, mostly in urban area(including our study area), can defend a 1000-y flood (based on the frequency analysis undertaken in 1984)( Yin et al,. 2013). Therefore,

100-y flooding can be well defended by flood wall and **1000-y flooding could be more**

**representative.**

(2)We tried to redefine the Pj parameter as you suggested, however, the simulation area of 'water depth <= 30 cm'(Figure 4) in the extreme scenario(1000-y flooding in the 2050s) was **too small**, few potential stations are in this area, which is not conducive to further analysis.

[Figure]

Fig.4 Inundation scenario of 1000-y flooding in the 2050s

**14. Line 355: The calculation of the OD matrix with the ArcGIS Network Analyst extension does not take into account the traffic jam? (see Line 192)**

Thanks for noting this. Although congestion data could be implemented into the modelling framework based on historic traffic data, we didn't use the congestion data due to uncertainties associated with how human behavior and patterns of congestion may differ under flood conditions when compared to normal conditions on which the traffic data were based. Furthermore, emergency vehicle lanes can also supply emergency vehicles in some roads so that unless road facilities are damaged in the case of disasters, emergency vehicles can drive at the maximum speed permitted by the road conditions.

To simulate the traffic jam during disaster, we considered that use the Risk Level to express the traffic jam during the crisis, we assumed that high risk level could bring heavy traffic jam, it could on behalf of the difficulties of the rescue.

So in this paper we used different road speed limit based on the People's Republic of China Technical Standard of Highway Engineering (JTG B01-2003) as the max speed to calculate the OD matrix.

**15. Line 374: "…i.e., the larger the service area, the larger the number of people who can be served by this station": this statement (affirmation) is not always true.**

Thanks for your comments. Yes, we know that the service area cannot replace the number of people who can be served by this station, we will modify the expression. But it is undeniable that the service area is also an important indicator of service capacity evaluation of an emergency rescue station. Many researches use service area to evaluate emergency responder accessibility. So in this paper we both used the service area and the served population as the judgment criteria to compare the service capacity of stations. In fact, in our study we have compared the difference of service area and the service population (Table2), we can see that in most case in our study area, larger service can serve more people.

Tab.2 Comparisons of service capacity under different disaster scenarios

| Scenarios | Response time(min) | Current service area(km$^2$) | Optimal service area(km$^2$) | current service population | optimal service population |
|---|---|---|---|---|---|
| no flood | 0-8 | 236.63 | 256.7 | 2088905 | 2174649 |
| | 0-12 | 300.52 | 306.8 | 2318052 | 2334324 |
| | 0-15 | 318.59 | 314.9 | 2368158 | 2356228 |
| 1000-year flood | 0-8 | 205.66 | 236.44 | 1838621 | 2081456 |
| in 2050 | 0-12 | 265.97 | 279.7 | 2186255 | 2213578 |
| | 0-15 | 282.93 | 286.52 | 2221628 | 2228562 |

[Figure]

Fig.5 Service capacity comparison with line chart

**16. Finally, what about the indirect impact of flooding (indirect vulnerability)?**

Apart from causing casualties, flooding may also damage emergency facilities(Figure 2); furthermore, flood inundation could damage to buildings and roads will lead to traffic congestion and render emergency rescue more difficult than usual, making rescue more difficult than usual and delaying the emergency response. The surface water flooding was shown to cause more disruption to emergency responders operating within the city due to its widespread and spatially distributed footprint when compared to fluvial flood events of comparable magnitude (Green et al., 2017).

**B. Questions related to the form of the article ("technical corrections")**

**1 Concerning the two selected flood scenarios (100 and 1000-years), what is the major historical flood that has been observed on the site? It would be interesting to consider the major historical hazard?**

Thanks for noting this. Our flooding scenarios results took reference by (Yin et al., 2011) research, and we choose two representative scenarios (100y or 1000y scenario)(Figure 2) to analysis the emergency response during flooding disasters.

As a typical tidal river, the Huangpu River is influenced by tides of the East China Sea with an average tide range of around 2.3 m at the river estuary. Given the low relief of the Huangpu River floodplain, it is subject to significant flood hazards from both the coastlines and the Huangpu River in the event of high tides. Indeed, the study area was frequently inundated by the Huangpu River until flood walls were erected during the 1950s. These have since been extended and reinforced. As a result, most of the study site is now protected from coastal and fluvial flooding, albeit again events with various return periods. Based on flood probability analysis carried out in 1984 by the Shanghai Water Authority, the design standard for flood walls along the Huangpu River was one in 1,000 years for urban area and one in 50 years in rural areas. However, our study area is rural area, so we it can be attacked by fluvial flooding more easily. Whatever flooding may destroy this area easily.

**2 Line 188 : "... the disaster risk level $m_i/p_j$ of the demand points/potential facilities**

Thanks for noting this.

The sentence has been revised as follows:

"the disaster risk level of the demand points($m_i$) and potential facilities($p_j$)."

**3 Line 322: why the alpha and beta weights are the same (equal)?**

It's an important question, in line 204 we calculated coverage level by Eq (9)

$$Q_i = INT(\alpha A1_i + \beta A2_i + \cdots + \varepsilon An_i + 1)$$

Where $\alpha, \beta \cdots \varepsilon$ represent the weights of the different indicators, i.e., their relative contributions to the estimated demand. The weights can be determined according to the actual situation of the study area, in case study, we regarded the population and the historical EMS calls for help at each demand point as the influencing factors $A_1$ and $A_2$. Both factors are important, so we did not quantify the weights of each factor, set alpha and beta weights the same.

We have added the explanation on this:

"we regarded the population density and the historical EMS calls for help at each demand point as the influencing factors $A_1$ and $A_2$, respectively of the demand coverage level (using Eq. (9)) and used equal weights for the two factors as for a specially instance ($\alpha = \beta = 0.5 * 10$)"

**4 Line 290: "Figure 3 shows the impact on the area serviced by each station for the different flood scenarios.**

**Line 298: it is difficult to make visually and easily the link between the coloured curves and the legend (the name of each EMS).**

**At least, the order of the name of the stations (12 EMS) in the legend must be the same than**

**the order of the curves to improve the reading of this Fig.**

Thanks for your comments, sorry for the low readability of Figure 3 in paper,

The Figure 3 in paper has been revised as follows:

[Figure]

**5 Line 411: Fig. 9 Comparisons of the average coverage level**

**Figure 9 shows "coverage level" in REAL values (3.54, 3.74 etc.) and not in INTEGER values?**

**(See equation 9, page 5)**

Thanks for noting this. Sorry, we didn't have a clear distinction between average coverage and coverage level.

The description and the Figure has been revised as:

"We combined the service areas of all emergency stations and used the Spatial Join tool in ArcGIS

10.2 to calculate how many times every demand point would be covered in 8, 12, and 15 minutes during the no-flooding and the worst-case flooding scenarios. To compare precisely, we then compared the average values (Figure 11)."

[Figure]

(Fig. 11 Comparisons of the average coverage value)

**6 Finally, I propose to the authors (if possible) to design a logi-gram related to the developed**
**methodology and results. Please, see example of the Figure 2, page 689, Alaeddine et al., 2015.**
Thanks for your comments, we added a logi-gram as below:

[Figure]

**7 We appreciate all the other technical corrections comments, and changes have been made**
**accordingly.**

**Reference**

Green, D., Yu, D. P., Pattison, I., Wilby, R., Bosher, L., Patel, R., Thompson, P., Trowell, K., Draycon, J., Halse, M., Yang, L. L., and Ryley, T.: City-scale accessibility of emergency responders operating during flood events, Nat Hazard Earth Sys, 17, 1-16, 2017.

Yin, J., Yin, Z. E., Hu, X. M., Xu, S. Y., Wang, J., Li, Z. H., Zhong, H. D., and Gan, F. B.: Multiple scenario analyses forecasting the confounding impacts of sea level rise and tides from storm induced coastal flooding in the city of Shanghai, China, Environ Earth Sci, 63, 407-414, 2011.

Yin, J., Yu, D., Yin, Z., Wang, J., and Xu, S. J. C. C.: Modelling the combined impacts of sea-level rise and land subsidence on storm tides induced flooding of the Huangpu River in Shanghai, China, 119, 919-932, 10.1007/s10584-013-0749-9, 2013.

---

## Referee Comment (RC2) · Anonymous Referee #2 · 3 Nov 2019

**Paper nhess-2019-214 "Multi-coverage Optimal Location Model for Emergency Medical Services (EMS) facilities under various disaster scenarios: A case study of urban fluvial floods in the Minhang District of Shanghai, China"**

**Comments**
This study proposes a methodology for location of emergency medical services (ems) facilities under various disaster scenarios. In the introduction section, some background information and relevant literature are presented. The methodology is discussed afterwards. A case study is performed for emergency evacuation as a result of floods in the Minhang District of Shanghai, China. Overall, I think the study investigates an interesting subject and fits well with the scope of the journal. However, there are quite a few areas that have to be improved in the paper. Below please find more specific comments:

*Page 1 line 14: I suggest replacing "add valuable minutes to travel times" to "may significantly increase the total travel time".
*Page 1 line 22: Since EMS is defined as "Emergency medical services" in the abstract, I suggest using "Emergency medical services" in the keywords as well instead of just "Emergency medical service".
*Pages 1: The authors start the introduction section with a discussion regarding the importance of emergency services. I suggest including a broader discussion, highlighting potential consequences of disasters, importance of emergency evacuation and disaster preparedness, and the need for developing the methodologies that can improve both emergency services and emergency evacuation. In the discussion, I recommend acknowledging some relevant studies, including the following:

- Wilmot, C. and Mei, B., 2004. Comparison of alternative trip generation models for hurricane evacuation. Natural Hazards review 5 (4), 170-178.
- Dulebenets, M. A., Abioye, O. F., Ozguven, E. E., Moses, R., Boot, W. R., & Sando, T. 2019. Development of statistical models for improving efficiency of emergency evacuation in areas with vulnerable population. Reliability Engineering & System Safety, 182, 233-249.
- Xu, Z., Yang, X., Zhao, X., and Li, L., 2012. Differences in driving characteristics between normal and emergency situations and model of car-following behavior. Journal of Transportation Engineering 138, 1303-1313.
- Cheng, G., Wilmot, C., and Baker, E., 2013. Development of a time-dependent disaggregate hurricane evacuation destination choice model. Natural Hazards Review 14, 163-174.
- Dulebenets, M. A., Pasha, J., Abioye, O. F., Kavoosi, M., Ozguven, E. E., Moses, R., Boot, W. R., & Sando, T. 2019. Exact and heuristic solution algorithms for efficient emergency evacuation in areas with vulnerable populations. International Journal of Disaster Risk Reduction, 101114.
- Sadri, A., Ukkusuri, S., Murray-Tuite, P., and Gladwin, H., 2014. How to evacuate: model for understanding the routing strategies during hurricane evacuation. Journal of Transportation Engineering 140, 61-69.

\*Page 3: Towards the end of the introduction section, please briefly discuss the structure of the manuscript (what would be described in the next sections of the manuscript).

\*Page 3: It would be good to have a Figure in section 2.1, illustrating the problem of interest. This will help the readers visualizing the problem at hand.

\*Pages 4-5: There are some issues with the control of indexes in the mathematical model. For example, in constraint set (2) you have y_j but you are summing over i, which is incorrect. The summation should be over index j. In constraint set (4) indexes "i,j" are not controlled. I assume you are trying to enforce the following condition: $t\_ij \leq T \; \forall i \in I, j \in J$. Again, please check the entire model and make sure that all the issues associated with the control of indexes are fixed.

\*Pages 7-8: Did you develop Figures 1 and 2 yourself? If not, please provide a relevant reference.

\*Page 15: The conclusion section should be strengthened. The authors should clearly highlight limitations of this study and how they will be addressed in future research.

---

## Author Comment (AC2) · 7 Nov 2019

We sincerely thank Reviewer #2 for his/her careful review and constructive feedback and suggestions. We truly believe that the changes suggested by Referee #2 will enhance the quality of the manuscript. A point-by-point response is presented below.

1.  **\*Page 1 line 14: I suggest replacing "add valuable minutes to travel times" to "may significantly increase the total travel time".**

    Thanks for your suggestion, the sentence has been changed as
    "However, disasters increase the difficulty of rescue and may significantly increase the total travel time between dispatch and arrival."

2.  **\*Page 1 line 22: Since EMS is defined as "Emergency medical services" in the abstract, I suggest using "Emergency medical services" in the keywords as well instead of just "Emergency medical service".**

    Thanks for noting this. The key words has been replaced.

3.  **\*Pages 1: The authors start the introduction section with a discussion regarding the importance of emergency services. I suggest including a broader discussion, highlighting potential consequences of disasters, importance of emergency evacuation and disaster preparedness, and the need for developing the methodologies that can improve both emergency services and emergency evacuation. In the discussion, I recommend acknowledging some relevant studies, including the following:**

    Thanks for your comments, we have read the relevant papers and one of the included references were also added to the reference list as below:

    "The demands being placed upon emergency services often exceed the resources made available by governments(Liu et al., 2017) . Furthermore, disasters always take more time to respond to the EMS demands due to a very dense traffic flow along the rescue routes. A crash at the rescue route may block one or several lanes, which will further result in congestion, significantly delay the emergencies efficiency, and may ultimately result in casualties (Dulebenets et al., 2019). Therefore, the maintenance of efficiency and quality of emergency services during disasters is the key to emergency management."

4.  **\*Page 3: Towards the end of the introduction section, please briefly discuss the structure of the manuscript (what would be described in the next sections of the manuscript).**

    Thanks for noting this. We have added discussion about the structure of the manuscript of introduction section as follows:

    "In the following sections, we describe the problems that the model needs to solve and the design of the Optimal Location Model. We also take a case study of urban fluvial floods in the

 Minhang District of Shanghai, China to validate this model."

5. **\*Page 3: It would be good to have a Figure in section 2.1, illustrating the problem of**
**interest. This will help the readers visualizing the problem at hand.**
Thanks for your comments, we added one qualitative expression of problems and one logi-gram
to make it clearer as follows:

[Figure]

Fig.1 Description of Model problems

[Figure]

Fig.2 Multi-coverage Optimal Location selection logi-gram

6. **Pages 4-5: There are some issues with the control of indexes in the mathematical model.**
**For example, in constraint set (2) you have $y\_j$ but you are summing over i, which is**
**incorrect. The summation should be over index j. In constraint set (4) indexes "i,j" are not**
**controlled. I assume you are trying to enforce the following condition: $t\_ij \leq T \; \forall i \in I, j$**
**$\in J$. Again, please check the entire model and make sure that all the issues associated**
**with the control of indexes are fixed.**

Thanks for noting this. The formulas has been corrected

7. **\*Pages 7-8: Did you develop Figures 1 and 2 yourself? If not, please provide a relevant**
**reference.**

Yes, Figures 1 and 2 in manuscript were developed by ourselves, Figure 1 was illustrated by
ArcGIS 10.2 software and Figure 2 was simulated by Floodmap model and illustrated by
ArcGIS 10.2.

8. **\*Page 15: The conclusion section should be strengthened. The authors should clearly**
**highlight limitations of this study and how they will be addressed in future research.**

Thanks for your suggestion, we have added the limitations of this study and the future research
in conclusion:

The model also has some aspects that could to be improved in order to arrive at more robust
solutions. Firstly, in our case study, we did not have a quantitative assessment of the disasters
risk level on emergency response, we evaluated the disaster risk level only by the buffer
distance to disaster source area, which is subjective. Secondly, as we only analyzed in theory,
our model did not consider whether the terrain or other basic conditions were suitable for the
EMS facilities. The future studies will consider disaster risk factors such as the vulnerability
of buildings comprehensively, evaluate the level of disaster risk quantitatively, and take the
real terrain and construction cost of each potential point into full account.

Lastly, the location of urban emergency service facilities has always been the focus of urban
planning. Location selection should consider a variety of factors and the ability to respond to
disasters is also a key factor to consider, while in this paper, we divided the area into grids with
a cell size of 2 km * 2 km and assumed that every grid center point was a potential emergency
station, The division of grid will affect the efficiency of model running efficiency and the
accuracy of results. The smaller the scale, the higher the accuracy, but the greater the running
pressure. Therefore, in the future research, we will consider multi-scale division with variable
squared meshes taking into account the distribution density of the population.

**Reference**

Dulebenets, M. A., Abioye, O. F., Ozguven, E. E., Moses, R., Boot, W. R., and Sando, T.:
Development of statistical models for improving efficiency of emergency evacuation in areas
with vulnerable population, Reliability Engineering & System Safety, 182, 233-249,
https://doi.org/10.1016/j.ress.2018.09.021, 2019.